# Feasibility of a Culturally Specific DEmentia Competence Education for Nursing Home Taskforce (DECENT) Programme: A Mixed-Method Approach

**DOI:** 10.3390/ijerph192416679

**Published:** 2022-12-12

**Authors:** Yayi Zhao, Yaping Ding, Li Liu, Helen Y. L. Chan

**Affiliations:** 1School of Nursing, Nanjing University of Chinese Medicine, No. 138 Xianlin Avenue, Xianlin District, Nanjing 210038, China; 2School of Nursing, Nanjing Medical University, No. 101 Longmian Avenue, Jiangning District, Nanjing 211100, China; 3Xiangya Nursing School, Central South University, No. 172 Tongzi Slopes Road, Yuelu District, Changsha 410013, China; 4The Nethersole School of Nursing, Faculty of Medicine, The Chinese University of Hong Kong, Sha Tin 999077, Hong Kong SAR, China

**Keywords:** competence, dementia, development, education, feasibility

## Abstract

Although educational resources have been developed to build staff‘s dementia care competence in Western culture, their applicability and cultural relevance to the Chinese population are questionable. To address this gap, the DEmentia Competence Education for Nursing home Taskforce (DECENT) programme was developed and tailored to Chinese staff. This study aimed to evaluate the feasibility and preliminary effects of the DECENT programme. A one-group pretest–posttest study, embedded with a qualitative component, was conducted among 12 healthcare professionals. The DECENT programme comprised eight topics covering essential competencies in dementia care. It was delivered face-to-face by a qualified educator once per week for 60–90 min over 8 weeks. Outcomes were measured at baseline and immediately post-intervention. A satisfaction survey and individual interviews were conducted post-intervention to understand participants’ perceptions and experience with the intervention. Nine participants finished the post-intervention assessment. Except for staff’s attitudes towards people with dementia, quantitative findings generally demonstrated positive changes following the intervention. Three categories were identified from the qualitative data: well-developed programme, perceived benefits, and barriers. The findings showed that the DECENT programme is feasible and is perceived by nursing home staff as relevant and useful to daily practice. A larger-scale study is needed to evaluate its effectiveness.

## 1. Introduction

The competence of nursing home staff in dementia care plays a pivotal role in the wellbeing of residents with dementia. Competence in nursing care is identified as focusing on the possession of discipline-related knowledge and skills, ability to apply sound judgement and skills and knowledge in professional practice, development of positive interpersonal relationships, and evaluation of outcome (e.g., quality of care) by standards [1]. Higher levels of staff competence have been proven to be associated with positive care outcomes in people with dementia, such as decreased agitation and aggression, decreased use of physical restraints and antipsychotics, and increased quality of life in people with dementia [2,3,4], as well as staff outcomes, such as lower levels of caregiving burden and higher levels of job satisfaction [5,6]. Conversely, poor dementia care knowledge and skills amongst care providers (e.g., lack of understanding about the challenging behaviours exhibited by people with dementia) undermine the quality of care [4].

### 1.1. Competency-Based Education in Dementia Care

Staff education has been prioritised by the World Health Organization (WHO) for improving the quality of dementia care [7]. Competency-based education is an approach that is organised around the related competencies needed in professional practice and allows learners to demonstrate their abilities in practice [8]. Competency-based educational programmes in dementia care have been developed in Australia and the United States, and positive effects on staff outcomes have been shown, including the self-rated level of confidence in overall caring and managing various caregiving tasks, understanding dementia and ability in applying person-centred care, improved knowledge, attitudes and self-efficacy related to dementia and dementia-related behaviours, and strategies in prevention and management of challenging behaviours [9,10,11].

Competency frameworks, which contain essential competencies for the workforce in dementia care, are used to guide the development of competency-based educational programmes in dementia care. Three competency frameworks in dementia care have been developed to guide the focus of educational programmes across care settings and disciplines [12,13,14]. These frameworks cover wide-ranging competencies, including knowledge and understanding of dementia and related challenging behaviours; physical and psychosocial wellbeing of people with dementia; management strategies; person-centred care; environment; interaction with families; end-of-life care; daily living activities; communication; care staff’s self-care and development; teamwork; and ethics in dementia care and therapeutic work.

Apart from the content, pedagogical methods affect the effectiveness of learning. Didactic teaching only is inadequate for improving staff competence and confidence in dementia care [15]. Competency-based education stresses that teaching strategies should be closely related to reality’s context, such as associating learners’ experience and real context with the education content, thereby promoting reflective learning and peer learning [16]. These strategies are consistent with adult-learning principles, such as promoting group interaction, facilitating relevancy-oriented learning, and connecting existing experiences and knowledge with new knowledge [17].

### 1.2. Development of a Culturally Specific Dementia Care Education in China

At present, an estimated 9.5 million people live with dementia [18,19], and approximately 44.5% to 56.4% residents in nursing homes live with dementia in China [20,21]. However, nursing home staff in mainland China generally have poor dementia knowledge [22], and the current care practice is dominated by medical-oriented care culture and punitive approaches towards challenging behaviours [23]. Despite the substantial demand for quality dementia care in nursing homes, relevant care competencies are being neglected by existing staff training [23]. Many nursing homes deny admission to people with dementia or discharge their residents if they develop dementia owing to the poorly equipped workforce [24]. Therefore, an educational programme to improve Chinese nursing home staff’s preparedness in dementia care is urgently needed.

Given cultural factors, such as culture-associated beliefs towards dementia, health policies and living experiences, may affect dementia care [25]. A great demand for educational intervention on dementia care tailored to the sociocultural context in low- and middle-income countries is recommended by the WHO. Culturally specific education is proven to be effective in increasing nurses’ attitudes towards dementia, knowledge, and dementia care [26]. Thus, the applicability of staff educational programmes and materials in dementia care developed in the Western countries is questionable in China owing to cultural differences in beliefs and lifestyles [24,27,28,29]. For example, filial piety from the Confucianism plays a crucial role in dementia care providers’ way of caring and interaction with people with dementia [30]; and a study has shown that Chinese residents favour Tang poems, rather than the newspapers [31]. To fill this gap, a culturally specific dementia care education, namely, DEmentia Competence Education for Nursing home Taskforce (DECENT) programme, was developed based on the Medical Research Council framework for developing and evaluating complex interventions [32]. The content of the DECENT programme (Appendix A) was obtained from research evidence on staff educational programmes for dementia care in nursing home settings through a systematic review [33], competency frameworks for dementia care workforce, and current practice gaps and training needs identified through an exploratory qualitative study on nursing homes in China [23].

The present study aimed to evaluate the feasibility of the DECENT programme amongst nursing home staff in mainland China. The objectives were to estimate the recruitment and retention rate, to test the feasibility of the intervention protocol and questionnaires, to evaluate the intervention adherence, to estimate the preliminary effects of the DECENT programme, and to understand participants’ perceptions and experience with the intervention.

## 2. Materials and Methods

### 2.1. Study Design

A one-group pretest–posttest study with an embedded qualitative component was conducted from May to July 2020. This study used the Standards for QUality Improvement Reporting Excellence (version 2.0) for reporting [34] (Appendix A).

### 2.2. Setting and Participants

The study was conducted in a private, 50-bed nursing home in Nanjing, mainland China. In this nursing home, around one-third of residents were living with dementia. Staff who met the following inclusion criteria were recruited: (a) involved in care for residents with dementia; (b) able to speak and read Chinese; and (c) willing to participate in the study. Those who had received any educational activities related to dementia in the recent three months before this study or planned to undergo relevant training throughout the entire study period were excluded.

### 2.3. Intervention

The DECENT programme comprises eight topics: ‘Understanding dementia and dementia related issues’, ‘Person-centred care’, ‘Care communication’, ‘Understanding challenging behaviours and management skills’, ‘Dementia care in daily living activities’, ‘Dementia-friendly environment’, ‘Interaction with families’, and ‘Care staff’s self-care and development’. Multiple pedagogical strategies were adopted, such as lectures, case study, video clips, reflection, and discussion (Appendix A).

The programme was delivered face-to-face through eight weekly sessions, with 60–90 min per session. Handouts for each session were distributed to participants in advance to facilitate self-directed learning. Homework with a format for self-reflection or case study related to the content of each session was given to participants after each session for practice. The education was delivered by the principal researcher who was experienced in nursing home staff training and participated in a train-the-trainer programme of a dementia care capacity building project in Hong Kong [31]. A research assistant with a master’s degree in geriatric nursing helped facilitate the teaching activities. During the study period, the research team provided ongoing support and consultation through a telecommunication platform (i.e., WeChat).

To ensure cultural sensitivity of the programme, the following strategies were used. First, practice gaps identified in our previous exploratory qualitative study were covered in the DECENT programme to meet the local needs for dementia care education [23]. This qualitative study aimed to explore the current practice for dementia care in nursing homes in mainland China and the training needs in dementia care of nursing home staff. The results showed that the dementia care practices were suboptimal, such as the hospital-like layout care environment, medical-oriented care culture, overlooking individual uniqueness and privacy, treating people with dementia as children, being authoritative, questing for culturally specific training and practices [23]. Second, the content of each topic was closely related to the nursing home context and common beliefs related to dementia in the Chinese community. All case scenarios used in the DECENT programme were adapted from the qualitative study to enhance familiarity and thus promote understanding. Third, the format and dosage were designed by considering the nursing home staff’s experience and working schedule to facilitate their adoption of the DECENT programme and to fit the educational programme into their routine schedule. From previous interviews, generally, one hour per session with a one-week interval was acceptable. The staff had difficulty participating in more frequent sessions. Moreover, the schedule should be arranged on the basis of the staff’s working schedule to fit the manpower arrangement and staff’s availability for training.

### 2.4. Measures

The focus of this study was the feasibility and acceptability of the intervention and questionnaires. Participants’ demographic characteristics including gender, age, education level, position, years working at the nursing home, and years working for people with dementia were obtained to gain insights into the acceptability of this study topic. Recruitment and retention rates were assessed by recording the number of potential participants in this nursing home and those maintained in the intervention. Intervention adherence was assessed by the attendance rate. Perception with the intervention was evaluated based on a 5-item self-developed satisfaction survey and qualitative interviews. Participants were asked to rate their level of satisfaction regarding the educational content, teaching format, trainer, usefulness, and overall evaluation of the DECENT programme. The responses were 5-point Likert from ‘very satisfied’ to ‘very dissatisfied’, scored from 5 to 1, respectively. They could also provide comments in an open-ended question.

Participants were also asked to complete the Chinese validated instruments used for outcome evaluation at baseline and immediately after the intervention. These instruments were as follows: Sense of Competence in Dementia Care Staff (SCIDS) scale [35,36] to measure staff’s sense of competence in dementia care; Dementia Knowledge Assessment Scale (DKAS) to assess staff’s dementia knowledge [22]; Approaches to Dementia Questionnaire (ADQ) [37] for assessing staff’s attitude towards people with dementia; person-centred care assessment tools (P-CAT) to assess the extent of person-centred care in a nursing home [38]; and the Neuropsychiatric inventory nursing home version (NPI-NH) [39,40] to assess the severity of behavioural and psychological symptoms of dementia (BPSD) of residents with dementia and staff-perceived disturbance. SCIDS, DKAS, ADQ, and P-CAT were self-administered questionnaires. If they needed any help, research assistants who were trained in outcome assessment would be immediately offered. To complete the NPI-NH, participants were asked to choose one of the residents with dementia under his/her care. Then the outcome assessor interviewed the participants about the BPSD of the resident with dementia and their perceived disturbance because of the BPSD.

### 2.5. Process Evaluation

Face-to-face interviews were conducted within one week after the intervention by the principal researcher to explore participants’ perceptions of and experience with the intervention. A purposive sample of participants with different ranks (including care assistants and nurses) was invited to maximise the sample variation. The sampled participants for the interviews were approached by the principal researcher to ask for their permission. A semi-structured interview guide with open-ended questions was used to guide the interviews. The major questions were ‘Please share your view about this educational programme’, ‘Which aspect(s) you like most? Why?’, and ‘How the programme can be improved?’.

### 2.6. Procedures

Permission for conducting the study was obtained from the nursing home manager. A nurse in the nursing home was trained to support the recruitment. She explained the study purpose and nature, as well as the voluntary nature of the participation, to the potential participants and obtained their written informed consent. The intervention schedule was decided mutually with the nursing home manager to facilitate staff participation. It was scheduled during the lunch break. After the recruitment, baseline assessment and intervention were performed in a designated meeting room separate from the living areas for residents. All participants and researchers followed infection-control measures (e.g., wearing masks and sterilising hands and all materials) because of the Coronavirus disease pandemic.

### 2.7. Ethical Considerations

This study was approved by The Joint Chinese University of Hong Kong New Territories East Cluster Clinical Research Ethics Committee (CREC Ref. No: 2019.500) and was registered in ClinicalTrials.gov (No: NCT04445077). This study was in accordance with the Declaration of Helsinki. After screening the eligibility of participants, an information sheet, which included the study information and the ethical issues of the research, was given to each of the participants. The participants were told that their involvement was voluntary; they could determine whether to participate or not and drop out at any time without any consequences. To ensure the participants’ full understanding, the study’s purpose and nature were informed again before the researchers conducted the baseline assessment. Before the baseline assessment, written informed consent was obtained from both the management of the nursing homes and all the participants. All the data were assigned a password to ensure security when stored in a computer. The field diaries documented by the principal investigator during the interviews and by the research assistant during the intervention were also locked in a drawer in the office. Moreover, only the researchers had access to the data. No publication kept any personal information of the participants.

### 2.8. Data Analysis

Quantitative data were analysed using SPSS version 25.0 (IBM Corp., Armonk, NY, USA). Descriptive statistics of mean and standard deviation (SD) or frequency and percentage were performed where appropriate. Given that this was a feasibility study, the focus of the analysis was to identify whether the score changes were geared towards the anticipated direction rather than to examine the statistical significance pre- and post-intervention [41].

All interviews were audiotaped and transcribed verbatim for qualitative content analysis as described by Graneheim and Lundman [42]. The verbatim transcripts were read through by the first author several times to obtain a whole sense. Text related to participants’ perception of the intervention were then extracted and condensed into meaning units and subsequently abstracted and labelled with codes. The first author and third author carried out this process separately, and then discussed the results and reached the consensus. The codes were compared and sorted into sub-categories and categories by identifying commonalities and variations. Finally, the fourth author checked all the coding and results. NVivo (version 12.0) was used to assist with data management.

## 3. Results

### 3.1. Recruitment and Retention

Among the 16 dementia care staff approached in this nursing home, 12 consented to participate in the study, giving a recruitment rate of 75%. The reasons for decline to participate were a busy working schedule, vacation, and study outside. Three of them withdrew from the study, resulting in a retention rate of 75%. One nurse and two care assistants quit their jobs before the completion of the intervention. The nurse manger reflected that ‘*this turnover rate was quite normal and was equal to the other period of time in the nursing home because the turnover rate here has always been high*’. After finishing the intervention, six participants were invited to participate in the interviews, but one care assistant refused because of the busy working schedule.

### 3.2. Participants’ Characteristics

Twelve staff members participated in the study, with seven care assistants (58.3%) and five nurses (41.7%). Two thirds of them were female (*n* = 8, 66.7%). Their median age was 52 years old (range = 25 to 58). The educational level of the care assistants was generally of junior high school or below, whereas those of nurses ranged from junior college (post-secondary nursing school training) to bachelor’s degree. Their median work experience in nursing homes and experience in dementia care was both two years, ranging within 1–16 and 1–10 years, respectively.

### 3.3. Intervention Adherence

Nine participants completed the intervention (completed at least four sessions and the post-intervention measures). Their average attendance rate was 86.1%, ranging from 50% to 100%. The mean completed sessions per participant was 6.9.

### 3.4. Preliminary Effects

No data were missing in the completed questionnaires. All participants could complete the DKAS, ADQ, P-CAT, and satisfactory scale by themselves within 40 min. NPI-NH was completed by research assistants by interviewing the participants about the BPSD of residents and their perceived disturbance within 20 min. Except for ADQ, a positive change occurred in the total scores of SCIDS, DKAS, and P-CAT. A decreasing trend was further observed in the severity of the BPSD and staff-perceived disturbance immediately post-intervention compared with the baseline (Table 1).

### 3.5. Participants’ Perception towards the Intervention

Participants answered satisfactory or very satisfactory in all items in the satisfaction scale. The mean scores of satisfactions with the educational content, the way in which the DECENT programme delivered, the trainer, and the programme usefulness were 4.44 (SD = 0.53), 4.22 (SD = 0.44), 4.22 (SD = 0.44), and 4.33 (SD = 0.50), respectively. The overall evaluation on the DECENT programme was 4.22 (SD = 0.44) out 5.00.

Apart from the quantitative results above, the results of interviews with three care assistants and two nurses were also reflected in their perceptions towards the intervention. The interviewees’ characteristics are shown in Table 2. Three categories were identified from the qualitative findings: well-developed programme, perceived benefits, and barriers (Table 3).

#### 3.5.1. Well-Developed Programme

**Content.** The participants reported that the content of the DECENT programme was comprehensive and highly relevant to their work, as one nurse stated:
*The training was closely related to our work in caring for residents with dementia. I can see your efforts (in designing this training). In our daily work, we often encounter similar situations mentioned in the training*.(P3, Nurse)

Care assistants also expressed their impressive topics, such as interaction with residents with dementia and families, caring for activities of daily life, and environment design.
*Pedagogical strategies. Consistent with the satisfaction survey, the participants endorsed the use of various teaching strategies during the interview. For example, case studies were appraised as ‘really helpful in connecting the training and their real experiences’*.(P3, Nurse)

**Trainer.** The trainer was appraised as qualified in this training by the participants. As one care assistant said:
*The lecturer was already good in performing this training. We can understand by following your thought and explanation in the class*.(P2, Care assistant)

#### 3.5.2. Perceived Benefits

**Gaining new perspectives towards dementia.** Some participants found that the intervention clarified the concept of dementia and introduced a new way of understanding the behaviours of the residents with dementia. Following the intervention, they attempted to appreciate the uniqueness of each resident and looked for the reasons behind these behaviours after training. A care assistant stated that,
*I am interested in this training because it is helpful for taking caring of residents with dementia. We never took any training on this aspect before. This is the first time*.(P1, Care assistant)

Another participant also shared,
*This programme provided us with great knowledge and skills that we did not have before. For example, I used to believe that dementia is a mental disorder, that they lost their mind. You could not talk to them before, but now I found that I can communicate with them. Sometimes, the more frequent I talk to them, the more I become interested in communicating. After all, they are cooperative, which has never happened before*.(P2, Care assistant)

**Applicability in practice.** Participants shared their positive experience of applying the knowledge they gained from the intervention into their daily practice in caring for residents with dementia (e.g., exploring the residents’ interests or communicating with them) to manage their challenging behaviours. For example, the nurses shared:
*A granny did not allow others to enter her room as she worried that her things would be stolen. She always sat in front of her room door to ensure security and did not attend any group activities. Now, we explain the reason for entering her room to her [rather than just instructing her what to do], invite her to join us, and lock the door for her. Most of the time, she understands and cooperates with us*.(P3, Nurse)
*We noted that the care assistants have tried to apply the knowledge and skills learned from the programme. For example, a granny always attempts to go home every afternoon. Previously, the care assistant only persuaded her to stay and followed her to ensure her safety, but these methods were ineffective. After the training, they have tried to engage her in games that she is interested in, and she is now less likely to request going home*.(P4, Nurse)

#### 3.5.3. Barriers

**Work tension.** The intervention schedule was fitted to staff’s working schedule to make them available for attending the training programme. However, some participants expressed that they were sleepy during the intervention because they were tired in their daily work and had a habit of taking a nap after lunch. Therefore, afterwards, when deciding the schedule for the intervention with managers from participating nursing homes, the factors that may affect learning effects such as resting habit would be taken into consideration.

**Challenges in learning.** Despite the learning motivation, the participants shared the challenges of learning new knowledge at the initial phase because they have never undergone compressive learning before. As a care assistants said:
*At the beginning, I could not remember what you taught. After all, we are in our 50s or 60s and we had never been through a comprehensive training like this before. However, during the second or third class, I felt that I could get used to the training. We still needed to learn the materials after classes.*(P2, Care assistant)

Therefore, hand-held cue cards with key points were designed for each topic to facilitate learning.

Participants generally found some teaching strategies, such as case studies and video clips, more interesting than didactic teaching. A nurse shared,
*Some other methods, such as case studies and watching video clips were better than didactic teaching during the training for care assistants*.(P4, Nurse)

To increase the learners’ interest, theoretical teaching was reduced, and the teaching formats focused on illustrating examples to facilitate understanding.

## 4. Discussion

A competency-based, culturally sensitive educational programme in dementia care, the DECENT programme, was developed for nursing home staff in mainland China based on the guideline from Medical Research Council framework [32]. The findings suggested that this programme was feasible and acceptable to participants, with potentially positive effects on their sense of competence, knowledge and care practice related to dementia care. Given current study was a feasibility study, the effects of the DECENT programme will be needed to be evaluated further in a study with a large sample size.

The attrition rate was 25% in this study, which was comparable with previous studies on educational programmes [43,44,45], while it was slightly lower than some others [46,47], and the most common reason was resigning. This result implies the reasonable retention rate of implementing the DECENT programme in nursing homes. However, the participants were from one small-scale nursing home and the sample size was small in this study, the attrition rate may have the occasionality. Thus, further study with large sample size is needed.

In China, professional dementia care has not been well provided in nursing homes. Consequently, care staff, including healthcare professionals and care assistants, lack relevant caregiving knowledge and skills [29]. For example, nursing home staff only scored 29.8 out of 50 in the Dementia Knowledge Scale [22]. Therefore, care staff have expressed difficulties in managing the BPSD of people with dementia, thus further resulting in huge caregiver burden and a high level of working strain [48]. Care assistants account for the majority of staff in nursing homes, but with education level in junior high school or lower and in their 50s [49]. This is also similar to the participants in this study. Thus, traditionally, they were given few training opportunities, especially in dementia care [23]. However, the results of this study have evidenced that the DECENT programme was feasible among Chinese care assistants in nursing homes.

The DECENT programme presented several strengths. First, it was developed based on existing evidence of dementia care educational programme and competency frameworks for dementia care in the literature. All essential dementia care competencies were included in the intervention [12,13,14]. The dose and format of the intervention were consistent with evidence of effective dementia education for the healthcare workforce [50]. Second, contextual factors related to local sociocultural factors and knowledge and practice gaps amongst nursing home staff were studied using a mixed-method approach [23] and were considered in the development process to enhance the cultural relevancy of the programme content. Cases used in the programme were derived from real-world experiences gained through observation and interviews in a qualitative study in four nursing homes in China to enhance participants’ resonance and interests [51]. Given the low educational level of care assistants, the teaching materials were adjusted to the appropriate reading level. Given that the use of jargon can hardly be avoided, explanations were provided. Third, the intervention was delivered through multiple pedagogical formats. Apart from didactic training, participants were encouraged to reflect upon their experience and discuss how new knowledge can be applied and enhance peer support. The intervention implementation was supported by the nursing home management so it could be flexibly fitted into the work schedule.

Drawing from the intervention implementation experience in this feasibility study, the DECENT programme was improved by designing hand-held cue cards with the important points of every topic to facilitate learning. To enhance participants’ learning interests, the didactic teaching was reduced. Instead, eased formats, such as case studies or video clips, were used to deliver the knowledge. When deciding the schedule for the intervention with managers from the participating nursing homes, the factors which had the possibility of affecting the learning effects, such as their resting habits, were taken into consideration.

We acknowledge several limitations of this feasibility study. First, the study clashed with the Coronavirus disease pandemic, so only one nursing home was recruited. Nevertheless, even though the sample size was small, the experience of the implementation procedure and the qualitative data revealed the feasibility of the intervention and thus provided insights for improvement before the main study. Another limitation of the study was that because of the practice issues (Coronavirus disease pandemic), direct observations on the staff’s competence in their daily work and response of residents with dementia could not be performed. Evaluation was based entirely on participants’ self-report outcomes, which could reflect the practice changes to some degree [52]. A larger-scale multisite trial using robust design is needed to evaluate the programme effects on staff competence and residents’ outcomes.

## 5. Conclusions

This study suggests that the DECENT programme is feasible and acceptable in nursing homes in mainland China, and with an acceptable recruitment and retention rate in the nursing home. It has the potential to fill the gaps in Chinese nursing home staff knowledge, their sense of competence in dementia care, and offers avenues for practice by equipping nursing home staff in dementia care, thereby improving quality of care. The DECENT programme is slightly modified based on the experience of this feasibility trial to enhance its efficacy for knowledge transfer. A larger-scale study will be conducted to evaluate its effectiveness.

## Figures and Tables

**Table 1 ijerph-19-16679-t001:** Outcome measures at pre- and post-intervention and the changes (N = 9) (mean ± SD).

	Pre-Intervention	Post-Intervention	Change (Post-Pre)
SCIDS ^1^	47.44 ± 6.39	49.44 ± 5.17	2.00 ± 7.66
Professionalism	15.67 ± 1.94	13.78 ± 2.28	−1.89 ± 2.89
Building relationships	10.89 ± 2.32	11.00 ± 2.60	0.11 ± 2.47
Care challenges	10.33 ± 1.50	12.22 ± 1.09	1.89 ± 1.54
Sustaining personhood	10.56 ± 2.01	12.44 ± 1.51	1.89 ± 2.42
DKAS ^1^	28.00 ± 13.11	30.11 ± 9.55	2.11 ± 8.34
Causes characteristics	6.67 ± 4.61	7.33 ± 3.74	0.67 ± 3.97
Risk and health promotion	6.67 ± 2.69	6.67 ± 2.65	−0.00 ± 1.66
Communication behaviour	7.11 ± 3.18	7.00 ± 3.16	−0.11 ± 2.47
Care consideration	7.56 ± 3.54	9.11 ± 1.96	1.56 ± 2.79
ADQ ^1^	64.78 ± 7.56	64.00 ± 6.87	−0.78 ± 4.58
Hope	23.56 ± 5.92	24.67 ± 3.32	1.11 ± 4.91
Person-centred	41.22 ± 4.21	39.33 ± 5.45	−1.89 ± 4.62
P_CAT ^1^	47.56 ± 5.00	52.11 ± 4.94	4.56 ± 7.72
Individualised care	21.89 ± 3.66	26.22 ± 2.91	4.33 ± 3.20
Organisational support	15.22 ± 4.66	14.56 ± 6.95	−0.67 ± 6.14
Environmental accessibility	10.44 ± 1.88	11.33 ± 0.50	0.89 ± 1.90
NPI_NH ^2^			
Severity of BPSD	43.78 ± 28.36	21.86 ± 8.13	−28.57 ± 25.44
Staff-perceived disturbance	16.67 ± 7.50	7.43 ± 4.20	−11.00 ± 5.89

Note. ^1^ total score ^2^ lower score represents better outcomes; otherwise higher score indicates better outcomes. ADQ = Approaches to Dementia Questionnaire. BPSD = behavioural and psychological symptoms of dementia. DKAS = Dementia Knowledge Assessment Scale. NPI-NH = Neuropsychiatric inventory nursing home version. P-CAT = person-centred care assessment tools. SCIDS = Sense of Competence in Dementia Care Staff scale.

**Table 2 ijerph-19-16679-t002:** Characteristics of participants involved in interviews in feasibility study.

Participant	Gender	Age	Education Level	Position	Experience of Working in Nursing Homes ^1^	Experience of Providing Dementia Care ^1^
P1	Female	58	Primary school	Care assistant	2	2
P2	Male	56	High school	Care assistant	2	2
P3	Female	27	College school	Nurse	1	1
P4	Female	37	Bachelor’s degree	Nurse	6	5
P5	Male	52	High school	Care assistant	2	2

Note. ^1^ year(s).

**Table 3 ijerph-19-16679-t003:** Overview of categories and sub-categories of feasibility study.

Categories	Sub-Categories
Well-developed programme	Content
	Pedagogical strategies
	Trainer
Perceived benefits	Gaining new perspective towards dementia
	Applicability in practice
Barriers	Work tension
	Challenges in comprehensive learning

## Data Availability

The data presented in this study are available on request from the corresponding author. The data are not publicly available due to ethical requirements.

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
