# Peer review of "Feasibility of a Culturally Specific DEmentia Competence Education for Nursing Home Taskforce (DECENT) Programme: A Mixed-Method Approach"

_ijerph, 2022, doi:10.3390/ijerph192416679_

Round 1

Reviewer 1 Report

This manuscript represents a study on the effects of the DECENT program on the Chinese nursing staff. The topic is pretty important to the dementia care field. More detailed studies are badly needed based on different cultures. Overall, I believe this work makes an important contribution to the field.

Comments:

1.     Authors highlight the cultural specificity of the Chinese. However, it’s hard to find the difference based on their DECENT outline.

2.     Their conclusion would be more convincing if they include more participants. More related works focusing on Chinses dementia care are needed in the discussion session.

Author Response

The point-by-point responses are in the box and are also attached as a Word file.

Response to Reviewer 1 Comments

Point 1: Authors highlight the cultural specificity of the Chinese. However, it’s hard to find the difference based on their DECENT outline.

Response 1: The cultural specificity of the Chinese context was highlighted in the main text (line 184-202). The cultural specificity was mainly reflected in explaining the content. For example, when educating the nursing-home staff how to manage challenging behaviours, the strategies which were accepted by Chinese older adults were introduced, such as reading Tang poems, singing revolutionary songs. In the outline of the DECENT programme, only the titles under each topic were displayed. To make it clearly, we highlighted some points in red in the outline which reflected the cultural specificity, such as “Approaches suggested to promote person-centred care in daily care for residents with dementia by taking the local context into consideration”, “Factors that commonly affect the communication with residents with dementia in Chinese nursing home settings”, “The potential reasons that may trigger challenging behaviours of residents with dementia in Chinese nursing-home settings”, “Commonly encountered problems and causes identified from the field visits and interviews with staff in nursing homes and literatures”.

Point 2: Their conclusion would be more convincing if they include more participants. More related works focusing on Chinses dementia care are needed in the discussion session.

Response 2: Thank you for the suggestions. Given this is a feasibility study, only one nursing-home staff were included to gain the implementation experience of the intervention. To evaluate the effects of the DECENT programme, a larger quasi-experimental study were conducted after this feasibility study. The quasi-experimental study was conducted within 10 Chinese nursing homes with 217 staff. Please refer to “Zhao Y*, Li L, Ding Y, Chan HYL. Effect of a culturally sensitive DEmentia Competence Education for Nursing home Taskforce (DECENT) programme in China: A quasi-experimental study. Nurse Education Today, 2022, 116:105434. https://doi.org/10.1016/j.nedt.2022.105434” for the results of the quasi-experimental study.

Chinese dementia care is briefly described in the discussion session (line 430-435). It was also introduced in the introduction session (line 76-85).

In the discussion session: “In China, professional dementia care has not been well provided in nursing homes. Consequently, care staff, including healthcare professionals and care assistants, lack relevant caregiving knowledge and skills [29]. For example, nursing-home staff only scored 29.8 out of 50 in the dementia knowledge scale [22]. Therefore, care staff have expressed difficulties in managing the BPSD of people with dementia, thus further resulting in huge caregiver burden and a high level of working strain [48].”

In the introduction session: “At present, an estimated 9.5 million people live with dementia [18, 19], and approximately 44.5% to 56.4% residents in nursing homes live with dementia in China [20, 21]. However, nursing-home staff in mainland China generally have poor dementia knowledge [22], and the current care practice is dominated by medical-oriented care culture and punitive approaches towards challenging behaviours [23]. Despite the substantial demand for quality dementia care in nursing homes, relevant care competencies are being neglected by existing staff training [23]. Many nursing homes deny admission to people with dementia or discharge their residents if they develop dementia owing to the poorly equipped workforce [24]. Therefore, an educational programme to improve Chinese nursing-home staff’s preparedness in dementia care is urgently needed.”

Reviewer 2 Report

The manuscript provides a strong rationale for the research need as the program the authors are testing fills a gap in culturally appropriate staff education in dementia care.

The following minor revisions are suggested to improve the impact of this research:

Literature review section: It is recommended that some additional content be added to the section "development of a culturally specific dementia-care education" that describes the research foundation for why culturally-specific education is needed, either broadly or in this area. What benefits does it provide specifically? What does research tell us in this area?

Under the "Intervention" section, around lines 143-145, a previous exploratory study is noted. Additional information about this should be provided. What was this study and what were the key gaps that were noted that informed the development of this work?

For the section "process evaluation" how were staff recruited for this portion of the evaluation? Along these lines, in section 3.1 Recruitment and retention-how many staff were approached to participate in the interviews? Did all who were approached agree to participate? How many ultimately provided feedback via the interviews?

Section 3.3, Intervention adherence, were there any trends noted with regard to topic and attendance? Also, what is your definition of "completion"? Did attendees need to complete all topic trainings to be considered "completers" for the purposes of this evaluation or was it only a certain portion of trainings?

Right before section 3.5.1 make it clear that you are now transitioning to qualitative findings by using a heading or some transition text. It wasn't clear to me as a reader that the content was switching from the survey into interview findings.

Discussion section: Add a sentence or two about the quantitative survey findings and that, while not the sole focus of this project, early data are trending in a direction that suggestions early content efficacy to be further explored at a later point in time.

The discussion section effectively summarizes the areas of strength for the program. However, more information on areas for improvements and modifications should also be discussed including the work tension, and learning challenges findings. How will you modify the program to address these areas? What implications do those barriers have for future educational efforts?

Author Response

Please see the point-by-point response to the comments below and it is also attached as a Word file.

Response to Reviewer 2 Comments

Point 1: Literature review section: It is recommended that some additional content be added to the section “development of a culturally specific dementia-care education” that describes the research foundation for why culturally-specific education is needed, either broadly or in this area. What benefits does it provide specifically? What does research tell us in this area?

Response 1: Thank you for the suggestion. The reason for culturally-specific education and the benefits of culturally-specific education are added (line 86-96).

“Given cultural factors, such as culture-associated beliefs towards dementia, health policies and living experiences, may affect dementia care[25]. A great demand for educational intervention on dementia care tailored to the socio-cultural context in low- and middle-income countries is recommended by the WHO. Culturally-specific education is proved to be effective in increasing nurses’ attitudes towards dementia, nowledge and dementia care [26]. Thus, the applicability of staff educational programmes and materials in dementia care developed in the Western countries is questionable in China owing to cultural differences in beliefs and lifestyles [24, 27–29]. For example, filial piety from the Confucianism plays a crucial role in dementia-care providers’ way of caring and interaction with people with dementia [30]; and study has shown that Chinese residents favour Tang poems, rather than the newspapers [31].”

Point 2: Under the "Intervention" section, around lines 143-145, a previous exploratory study is noted. Additional information about this should be provided. What was this study and what were the key gaps that were noted that informed the development of this work?

Response 2: The aims and the key findings (gaps of dementia care in Chinese nursing homes) were added (line 187-192).

“This qualitative study aimed to explore the current practice for dementia care in nursing homes in mainland China and the training needs in dementia care of nursing home staff. The results showed that the dementia care practices were suboptimal, such as the hospital-like layout care environment, medical-oriented care culture, overlooking individual uniqueness and privacy, treating people with dementia as children, being authoritative, questing for culturally specific training and practices [23].”

Point 3: For the section "process evaluation" how were staff recruited for this portion of the evaluation? Along these lines, in section 3.1 Recruitment and retention-how many staff were approached to participate in the interviews? Did all who were approached agree to participate? How many ultimately provided feedback via the interviews?

Response 3: In the section "process evaluation", the recruitment of the interviewees was added (line 237-238).

“The sampled participants for the interviews were approached by the principal researcher to ask for their permission.”

In section 3.1 Recruitment and retention, after finishing the intervention, six participants were invited to participate in the interviews, but one care assistant refused because of the busy working schedule. This statement was described (297-299).

“After finishing the intervention, six participants were invited to participate in the interviews, but one care assistant refused because of the busy working schedule.”

Point 4: Section 3.3, Intervention adherence, were there any trends noted with regard to topic and attendance? Also, what is your definition of “completion”? Did attendees need to complete all topic trainings to be considered “completers” for the purposes of this evaluation or was it only a certain portion of trainings?

Response 4: There was no trends noted with regard to the topic and attendance. Participants who completed at least four sessions and the post-intervention measures were considered as “completers”. This explanation was added (line 309-310).

“(completed at least four sessions and the post-intervention measures)”

Point 5: Right before section 3.5.1 make it clear that you are now transitioning to qualitative findings by using a heading or some transition text. It wasn't clear to me as a reader that the content was switching from the survey into interview findings.

Response 5: The transition text was added (line 332-333).

“Apart from the quantitative results above, the results of interviews with three care assistants and two nurses were also reflected their perceptions towards the intervention.”

Point 6: Discussion section: Add a sentence or two about the quantitative survey findings and that, while not the sole focus of this project, early data are trending in a direction that suggestions early content efficacy to be further explored at a later point in time.

Response 6: The quantitative findings were stated and the needs of further study to evaluate the effects of the DECENT programme is stated (line 418-422).

“The findings suggested that this programme was feasible and acceptable to participants, with potentially positive effects on their sense of competence, knowledge and care practice related to dementia care. Given current study was a feasibility study, the effects of the DECENT programme will be needed to be evaluated further in a study with a large sample size.”

Point 7: The discussion section effectively summarizes the areas of strength for the program. However, more information on areas for improvements and modifications should also be discussed including the work tension, and learning challenges findings. How will you modify the program to address these areas? What implications do those barriers have for future educational efforts?

Response 7: Based on the implementation experience of this feasibility study, the improvement and modifications of the intervention was made for further study. The details were explained in the discussion section (line 460-467).

“Drawing from the intervention implementation experience in this feasibility study, the DECENT programme was improved by designing hand-held cue cards with the important points of every topic to facilitate learning. To enhance participants’ learning interests, the didactic teaching was reduced. Instead, eased formats, such as case studies or video clips, were used to deliver the knowledge. When deciding the schedule for the intervention with managers from the participating nursing homes, the factors which had the possibility of affecting the learning effects, such as their resting habits, were taken into consideration.”
